

**Holocene dynamics in the Bering Strait inflow to the Arctic and the Beaufort Gyre**
**circulation based on sedimentary records from the Chukchi Sea**
Masanobu Yamamoto[1-3*], Seung-Il Nam[4], Leonid Polyak[5], Daisuke Kobayashi[3], Kenta
Suzuki[3], Tomohisa Irino[1,3], Koji Shimada[6]
*[1]Faculty of Environmental Earth Science, Hokkaido University, Kita-10, Nishi-5,*
*Kita-ku, Sapporo 060-0810 Japan*
*[2]Global Institution for Collaborative Research and Education, Hokkaido University,*
*Kita-10, Nishi-5, Kita-ku, Sapporo 060-0810 Japan*
*[3]Gradute School of Environmental Science, Hokkaido University, Kita-10, Nishi-5,*
*Kita-ku, Sapporo 060-0810 Japan*
*[4]Korea Polar Research Institute, 26 Songdomirae-ro, Yeonsu-gu, Incheon 21990,*
*Republic of Korea*
*[5]Byrd Polar and Climate Research Center, The Ohio State University, Columbus, OH*
*43210USA*
*[6]Tokyo University of Marine Science and Technology, 4-5-7, Konan, Minato-ku, Tokyo*
*108-8477, Japan.*
*\*Corresponding author. Tel: +81-11-706-2379, Fax: +81-11-706-4867, E-mail*
*address: myama@ees.hokudai.ac.jp (M. Yamamoto)*
**ABSTRACT**
The Beaufort Gyre (BG) and the Bering Strait inflow (BSI) are important elements of
the Arctic Ocean circulation system and major controls on the distribution of Arctic sea





ice. We report records of the quartz/feldspar and chlorite/illite ratios in three sediment
cores from the northern Chukchi Sea providing insights into the long-term dynamics of
the BG circulation and the BSI during the Holocene. The quartz/feldspar ratio, a proxy
of the BG strength, gradually decreased during the Holocene, suggesting a long-term
decline in the BG strength, consistent with orbitally-controlled decrease in summer
insolation. We suppose that the BG rotation weakened as a result of increasing stability
of sea-ice cover at the margins of the Canada Basin, driven by decreasing insolation.
Millennial to multi-centennial variability in the quartz/feldspar ratio (the BG
circulation) is consistent with fluctuations in solar irradiance, suggesting that solar
activity affected the BG strength on these timescales. The BSI approximated by the
chlorite/illite record shows intensified flow from the Bering Sea to the Arctic during the
middle Holocene, which is attributed primarily to the effect of an overall weaker
Aleutian Low. The middle Holocene intensification of the BSI was associated with
decrease in sea ice concentrations and increase in marine production, as indicated by
biomarker concentrations, suggesting an influence of the BSI on sea ice distribution and
biological production in the Chukchi Sea.
**1. Introduction**

The Arctic currently faces rapid climate change caused by global warming (e.g.,

Screen and Simmonds, 2010; Harada, 2016). Changes in the current system of the
Arctic Ocean regulate the state of Arctic sea ice and are involved in global processes via
ice albedo feedback and the delivery of freshwater to the North Atlantic Ocean (Miller
et al., 2010; Screen and Simmonds, 2010). The most significant consequence of this
climate change during recent decades is the retreat of summer sea ice in the Pacific



sector of the Arctic (e.g., Shimada et al., 2006; Harada et al., 2016, and references
therein). Inflow of warm Pacific water through the Bering Strait (hereafter Bering Strait
Inflow [BSI]) is suggested to have caused catastrophic changes in sea ice stability in the
western Arctic Ocean (Shimada et al., 2006). Comprehending these changes requires
investigation of a longer-term history of circulation in the western Arctic and its
relationship to atmospheric forcings. Within this context, the Chukchi Sea is a key
region to understand the western Arctic current system as it is located at the crossroads
of the BSI and the Beaufort Gyre (BG) circulation in the western Arctic Ocean (Fig. 1)
(e.g., Winsor and Chapman, 2004; Weingartner et al., 2005).
In this paper we apply mineralogical proxies of the BG and BSI to sediment cores
with a century-scale resolution from the northern margin of the Chukchi shelf. The
generated record provides new understanding of changes in the BG circulation and BSI
strength during most of the Holocene (last ~9 ka). We discuss the possible causes and
forcings of the BG and BSI variability, as well as its relationship to sea-ice history and
biological production in the western Arctic.

**2. Background information**
*2.1. Oceanographic settings*
The wind-driven surface current system of the Arctic Ocean consists of the BG and
the Transpolar Drift (TPD) (Proshutinsky and Johnson, 1997; Rigor et al., 2002). This
circulation is controlled by the atmospheric system known as the Arctic Oscillation
(AO) (Rigor et al., 2002). When the AO is in the positive phase, the BG shrinks back
into the Beaufort Sea, the TPD expands to the western Arctic Ocean, and the sea-ice
transport from the eastern Arctic to the Atlantic Ocean is intensified. When the AO is in



negative phase, the BG expands, the TPD is limited to the eastern Arctic, and sea ice is
exported efficiently from the Canada Basin to the eastern Arctic. Thus, sea-ice
distribution is closely related to the current system.

A dramatic strengthening of the BG circulation occurred during the last two decades

(Shimada et al., 2006; Giles et al., 2012). This change was attributed to a recent
reduction in sea-ice cover along the margin of the Canada Basin, which caused a more
efficient transfer of the wind momentum to the ice and underlying waters in the BG
(Shimada et al., 2006). The delayed development of sea ice in winter enhanced the
western branch of the Pacific Summer Water across the Chukchi Sea. This anomalous
heat flux into the western part of the Canada Basin retarded sea-ice formation during
winter, thus, further accelerating overall sea-ice reduction.

The BSI, an important carrier of heat and freshwater to the Arctic, transports the

Pacific water to and across the Chukchi Sea, interacts with the BG circulation at the
Chukchi shelf margin (e.g., Shimada et al., 2006). After passing the Bering Strait the
BSI flows in three major branches. One branch, the Alaskan Coastal Current (ACC),
runs northeastward along the Alaskan coast as a buoyancy-driven boundary current
(Red arrow in Fig. 1; Shimada et al., 2001; Pickart, 2004; Weingartner et al., 2005). The
second, central branch follows a seafloor depression between Herald and Hanna Shoals,
then turns eastward and merges with the ACC (Yellow arrow in Fig. 1; Winsor and
Chapman, 2004; Weingartner et al., 2005). The third branch flows northwestward,
especially when easterly winds prevent the ACC (Winsor and Chapman, 2004). This
branch may then turn eastward along the shelf break (Blue arrow in Fig. 1; Pickart et al.,

2010).



The BSI is driven by a northward dip in sea level between the North Pacific and the
Arctic Ocean (Shtokman, 1957; Coachman and Aagaard, 1966). There has been a
long-standing debate, whether this dipping is primarily controlled by steric difference
(Stigebrandt, 1984) or from wind-driven circulations (Gudkovitch, 1962). Stigebrandt
(1984) assumed that the salinity difference between the Pacific and Atlantic Oceans
causes the steric height difference between the Bering Sea and the Arctic Ocean.
Aagaard et al. (2006) argued that the local salinity in the northern Bering Sea controlled
the BSI, although wind can considerably modify the BSI on a seasonal timescale. De
Boer and Nof (2004) proposed a model that the mean sea level difference along the
strait is set up by the global winds, particularly the strong Subantarctic Westerlies.
Recently, a conceptual model of the BSI controls has been developed based on a
decade of oceanographic observations (Danielson et al., 2014). According to this model,
storms centered over the Bering Sea excite continental shelf waves on the eastern
Bering shelf that intensify the BSI on synoptic time scales, but the integrated effect of
these storms tends to decrease the BSI on annual to decadal time scales. At the same
time, an eastward shift and overall strengthening of the Aleutian Low pressure center
during the period between 2000–2005 and 2005–2011 increased the sea level pressure
in the Aleutian Basin south of the Bering Strait by 5 hPa, in contrast to overall
decreased pressure of the Aleutian Low system, thus decreasing the water column
density through isopycnal uplift by weaker Ekman suction. This change thereby raised
the dynamic sea surface height by 4.2 m along the Bering Strait pressure gradient,
resulting in the BSI increase by 4.5 cm/s, or 0.2 Sv (calculated based on the
cross-section area of $4.25 \times 10^{6}$ m$^{2}$). This increase constitutes about one quarter of the
average long-term BSI volume of ~0.8 Sv (Roach et al., 1995). Such a large



contribution clearly identifies changes in the Aleutian Low strength and position as a
key factor regulating the BSI on inter-annual time scales.
The BSI also transports nutrient from the Pacific to the Arctic. A rough estimation
suggests that the BSI waters significantly contribute to marine production in the Arctic
(Yamamoto-Kawai et al., 2006). High marine production in the Chukchi Sea of up to
400 gC m$^{-2}$ y$^{-1}$ in part is thought to reflect the high nutrient fluxes by the BSI (Walsh
and Dieterle, 1994; Sakshaug, 2004). A recent enhancement of biological productivity
and the biological pump in the Beaufort and Chukchi Seas has been associated with the
retreat of sea ice (summarized by Harada et al., 2016). This phenomenon is attributed to
an increase of irradiance in the water column (Frey et al., 2011; Lee and Whitledge,
2005), wind-induced mixing that replenishes sea surface nutrients (Carmack et al.,
2006), and their combination (Nishino et al., 2009). However, the nutrient flux into the
Arctic Ocean was not evaluated in this context. The investigation of BSI intensity and
marine production during the Holocene will be useful to understand on-going changes
in marine production in the Arctic Ocean.

***2.2. Mineral distribution in the Chukchi Sea sediments***
Spatial variation in mineral composition of surficial sediments along the western
Arctic margin has been investigated in a number of studies using different
methodological approaches but showing an overall consistent picture (e.g., Naidu et al.,
1982; Naidu and Mowatt, 1983; Wahsner et al., 1999; Kalinenko, 2001; Viscosi-Shirley
et al., 2003; Darby et al., 2011; Kobayashi et al., 2016). A recent study of mineral
distribution in sediments from the Chukchi Sea and adjacent areas of the Arctic Ocean
and the Bering Sea suggests that the quartz/feldspar (Q/F) ratio is higher on the North



American than on the Siberian side of the western Arctic (Fig. 2; Kobayashi et al.,
2016). These results are consistent with earlier studies including mineral determinations
of shelf sediments and adjacent coasts (Vogt, 1997; Stein, 2008; Darby et al., 2011). In
particular, data of Darby et al. (2011), although quantified by a different method, also
show a trend of decreasing Q/F ratio from North American margin to the Chukchi Sea
and further to the East Siberian Sea. This zonal gradient of the Q/F ratio suggests that
quartz-rich but feldspar-poor sediments are derived from the North American margin by
the BG circulation, whereas feldspar–rich sediments are delivered to the Chukchi Sea
from the Siberian margin by currents along the East Siberian slope (Kobayashi et al.,
2016). Thus, this ratio can be used as a provenance index for the BG circulation
reflecting changes in its intensity in sediment-core records (Kobayashi et al., 2016).
Kaolinite is generally minor clay in the western Arctic but relatively abundant in the
Northwind Ridge and Mackenzie Delta areas where the BG circulation exerts an
influence (Naidu and Mowatt, 1983; Kobayashi et al., 2016). Kaolinite in the
Northwind Ridge originated from ancient rocks exposed on the North Slope and was
delivered by water or sea ice via the Beaufort Gyre circulation (Kobayashi et al., 2016).
Kobayashi et al. (2016) also indicate that both the (chlorite + kaolinite)/illite and
chlorite/illite ratios (CK/I and C/I ratios, respectively) are higher in the Bering Sea and
decrease northward throughout the Chukchi Sea, reflecting the diminishing strength of
the BSI (Fig. 2). These results are consistent with earlier studies showing that illite is a
common clay mineral in Arctic sediments (Kalinenko, 2001; Darby et al., 2011),
whereas, chlorite is more abundant in the Bering Sea and the Chukchi shelf areas
influenced by the BSI (Naidu and Mowatt, 1983; Kalinenko, 2001; Nwaodua et al.,
2014; Kobayashi et al., 2016). Chlorite occurs abundantly near the Bering Sea coasts of



Alaska, Canada, and the Aleutian Islands (Griffin and Goldberg, 1963). The
chlorite/illite ratio is higher in the bed load of rivers and deltaic sediments from
southwestern Alaska than from northern Alaska and East Siberia, reflecting differences
in the geology of the drainage basins (Naidu and Mowatt, 1983). Because chlorite
grains are more mobile than illite grains under conditions of intense hydrodynamic
activity, chlorite grains are transported a long distance from the northern Bering Sea to
the Chukchi Sea via the Bering Strait (Kalinenko, 2001). In the surface sediments of the
Chukchi Sea, the CK/I ratio shows a good correlation with the C/I ratio, indicating that
both ratios can be used as a provenance index for the BSI (Kobayashi et al., 2016).
Ortiz et al. (2009) constructed the first chlorite-based Holocene record of the BSI by
quantifying the total chlorite plus muscovite abundance based on diffuse spectral
reflectance of sediments from a northeastern Chukchi Sea core. The record shows a
prominent intensification of the BSI in the middle Holocene. However, a record from
just one site is clearly insufficient to characterize sedimentation and circulation history
in such a complex area. More records of mineral proxy distribution covering various
oceanographic and depositional environments are needed to further our understanding
of the evolution of the BSI.
The Holocene dynamics of the BG circulation is also poorly understood. A study of
sediment core from the northeastern Chukchi slope identified centennial- to
millennial-scale variability in the occurrence of Siberian iron oxide grains presumably
delivered via the BG (Darby et al., 2012). However, transport of these grains depends
not only on the BG, but also on circulation and ice conditions in the Eurasian basin,
which complicates the interpretation and necessitates further proxy studies of the BG
history.






### 3. Samples and methods

This study uses three sediment cores from the northern and northeastern margins of
the Chukchi Sea: ARA02B 01A-GC (gravity core; 563 cm long; 73°37.89'N,
166°30.98'W), HLY0501-05JPC/TC (jumbo piston core/trigger; 1648 cm long,
72°41.68'N, 157°31.20'W) and HLY0501-06JPC (1554 cm long; 72°30.71'N,
157°02.08'W) collected from 111 m, 462 m and 673 water depth, respectively (Fig. 1).
The sediments in 01A-GC and in the Holocene part of 05JPC/TC (0–1300 cm) and
06JPC (0–935 cm) consist predominantly of homogeneous clayey silt (fine-grained
unit). This unit of cores 05JPC and 06JPC is underlain by a more complex
lithostratigraphy with laminations and coarse ice rafted debris indicative of
glaciomarine environments affected by glacial/deglacial processes ("glaciomarine unit";
McKay et al., 2008; Lisé-Pronovost et al., 2009; Polyak et al., 2009).
Age was constrained by seven accelerator mass spectrometry (AMS) $^{14}$C ages of
mollusc shells from core 01A-GC (Stein et al., 2017). The core-top of 01A-GC was
assumed to be sediment surface because labile organic compounds such as $IP_{25}$ and
sterols show a downcore decreasing trend in their concentrations in the top 10 cm (Stein
et al., 2017), which is commonly seen in ocean surface sediments, suggesting that the
lost of surface sediments was minimal during coring. $^{14}$C ages were converted to
calendar ages using the CALIB7.0 program and marine13 dataset (Reimer et al., 2013).
Local reservoir correction (ΔR) was assumed 500 years for 01A-GC (McNeely et al.,
2006; Darby et al., 2012).
In core 05JPC/TC, age was constrained by six AMS $^{14}$C ages of mollusc shells from
core 05JPC (Barletta et al., 2008; Darby et al., 2009). Local reservoir correction (ΔR)





was assumed 0 years for 05JPC (McNeely et al., 2006; Darby et al., 2012). Concurrent
age constraints for 05JPC were provided by $^{210}$Pb determinations in the upper part
(05TC) and paleomagnetic analysis (Barletta et al., 2008; McKay et al., 2008; Darby et
al., 2012). The age model of core 05JPC/TC was constructed by linear interpolation
between the $^{14}$C datings (2.4–7.7 ka) as well as the assumed modern age of the 05TC
top, with the assumption that the offset of JPC to TC is 75 cm (Polyak et al., 2016).
Ages below the dated range were extrapolated to the bottom of homogenous
fine-grained unit at 1300 cm (9.4 ka).

In core 06JPC, age was tentatively constrained by ten paleointensity datums based on

the $^{14}$C ages of nearby cores and a $^{14}$C age of benthic foraminifera (8.16 ka at 918 cm)
(Lisé-Pronovost et al., 2009), with the assumption that the offset of JPC to TC is 147
cm (Ortiz et al., 2009). The age model of core 06JPC was constructed by linear
interpolation between the paleointensity datums (2.0–7.9 ka).

In total 110 samples were collected for mineralogical analysis from core 01A-GC at

intervals averaging 5 cm (equivalent to approximately 80–90 years) down to a depth of
545 cm (ca. 9.3 ka). In core 05JPC/TC, 44 samples were collected from fine-grained
unit at intervals averaging 30 cm (equivalent to approximately 210–220 years) down to
a depth of 1286 cm (ca. 9.3 ka), and 7 samples were collected from the underlying
glaciomarine sediments. In core 06JPC, 79 samples were collected from fine-grained
unit at intervals of 10 cm (equivalent to approximately 90 years) down to a depth of 937
cm (ca. 8.0 ka), and 46 samples were collected from the underlying glaciomarine unit.

We also analyzed 16 surface sediment samples (0–1 cm) from the eastern Beaufort

Sea near Mackenzie delta and 3 surface sediment samples (0–1 cm) from the western
Beaufort Sea (Fig. 2) to fill the gaps in the dataset of Kobayashi et al. (2016). These



were obtained during the RV Araon cruises in 2013 and 2014 (ARA04C and ARA05C,
respectively; supplementary table 1).
Mineral composition was analyzed on MX-Labo X-ray diffractometer (XRD)
equipped with a CuKα tube and monochromator. The used tube voltage and current
were 40 kV and 20 mA, respectively. Scanning speed was 4°2θ/min and the data
sampling step was 0.02°2θ. Each powdered sample was mounted on a glass holder with
a random orientation and X-rayed from 2 to 40°2θ. An additional precise scan with a
scanning speed of 0.2°2θ/min and sampling step of 0.01°2θ from 24 to 27°2θ was
conducted to distinguish chlorite from kaolinite by evaluation of the peaks around
25.1°2θ (Elvelhøi and Rønningsland, 1978). In this study, the background-corrected
diagnostic peak intensity was used for evaluating the abundance of each mineral. The
relative XRD intensities of quartz at 26.6°2θ (d = 3.4 Å), feldspar including both
plagioclase and K-feldspar at 27.7°2θ (d = 3.2 Å), illite including mica at 8.8°2θ (d =
10.1 Å), chlorite including kaolinite (called "chlorite+kaolinite" hereafter) at 12.4°2θ (d
= 7.1 Å), kaolinite at 24.8 °2θ (d = 3.59 Å) and chlorite at 25.1°2θ (d = 3.54 Å) were
determined using MacDiff software (Petschick, 2000) based on the peak identification
protocols of Biscaye (1965).
The mineral ratios used in this study are defined based on XRD peak intensities (PI)
as:
Q/F = quartz/feldspar = [PI at 26.6°2θ]/[PI at 27.7°2θ]
CK/I = (chlorite+kaolinite)/illite = [PI at 12.4°2θ]/[PI at 8.8°2θ]
C/I = chlorite/illite = [PI at 25.1°2θ]/[PI at 8.8°2θ]
K/I = kaolinite/illite = [PI at 24.8°2θ]/[PI at 8.8°2θ]





The standard error of duplicate analyses in all samples averaged 1.1, 0.08 and 0.05
for Q/F, CK/I and C/I ratios, respectively.
Clay minerals (less than 2-μm diameter) in core 01A-GC were separated by the
settling method based on the Stokes' law (Müller, 1967). To produce an oriented powder
X-ray   diffractometry   (XRD)   sample,   the   collected   clay   suspensions   were
vacuum-filtered onto 0.45-μm nitrocellulose filters and dried. Ethylene glycol (50 μl)
was then soaked onto the oriented clay on the filters. Glycolated sample filters were
stored in an oven at 70°C for four hours and then immediately subjected to XRD
analyses. Each sample filter was placed directly on a glass slide and X-rayed with a tube
voltage of 40 kV and current of 20 mA. Scanning speed was 0.5°2θ/min and the
data-sampling step was 0.02°2θ from 2 to 15°2θ. An additional precise scan with a
scanning speed of 0.2°2θ/min and sampling step of 0.01°2θ from 24 to 27°2θ was
conducted to distinguish chlorite from kaolinite by evaluation of the peaks around
25.1°2θ (Elvelhøi and Rønningsland, 1978). The standard errors of duplicate analyses in
all samples averaged 0.05 and 0.06 for CK/I and C/I ratios, respectively.
The diffraction intensity of chlorite+kaolinite at 7.1 Å was significantly positively
correlated with that of chlorite at 3.54 Å ($r = 0.89$), but not with that of kaolinite at 3.59
Å ($r = 0.39$) in western Arctic surface sediments (Kobayashi et al., 2016), indicating that
the diffraction intensity of chlorite+kaolinite is governed by the amount of chlorite rather
than that of kaolinite.
Spectral analysis of the downcore Q/F and C/I variability was performed using the
maximum entropy method provided in the Analyseries software package (Paillard et al.,

1996).






**4. Results**
*4.1. Surface sediments of the Beaufort Sea*
Because the dataset of Kobayashi et al. (2016) has only one sample in the eastern
Beaufort Sea, we added the data of 16 samples from the eastern Beaufort Sea near the
Mackenzie delta and 3 samples from the western Beaufort Sea to fill the gaps in their
dataset. More clearly than Kobayashi et al. (2016), the new combined dataset shows that
the surface sediments in the eastern Beaufort Sea have the higher Q/F and lower CK/I
and C/I ratios than those in the Chukchi Sea (Fig. 2A–C; Supplementary table 1).
The Q/F ratio showed a westward decreasing trend from the eastern Beaufort Sea to
the East Siberian Sea and its offshore area (Fig. 2D). This supports a notion that
quartz-rich but feldspar-poor sediments are derived from the North American margin by
the BG circulation, whereas feldspar–rich sediments are delivered to the Chukchi Sea
from the Siberian margin by currents along the East Siberian slope (Vogt, 1997; Stein,
2008; Darby et al., 2011; Kobayashi et al., 2016).
The CK/I and C/I ratios showed a northward decreasing trend in the Chukchi Sea and
the Chukchi Borderland (Fig. 2E). This result are consistent with earlier studies
showing that illite is a common clay mineral in Arctic sediments (Kalinenko, 2001;
Darby et al., 2011), whereas, chlorite is more abundant in the Bering Sea and the
Chukchi shelf areas influenced by the BSI (Naidu and Mowatt, 1983; Kalinenko, 2001;
Nwaodua et al., 2014; Kobayashi et al., 2016).
These trends support the conclusion of Kobayashi et al. (2016) mentioning that the
Q/F ratio can be used as a provenance index for the BG circulation reflecting a
westward decrease in its intensity, and the CK/I and C/I ratios can be used as a
provenance index for the BSI reflecting a northward decrease in its intensity. The



provenance and transportation of these detrital minerals are discussed in detail in Naidu
and Mowatt (1983), Kalinenko (2001), Nwaodua et al. (2014) and Kobayashi et al.

(2016).


### *4.2. Cores 01A-GC, 05JPC/TC and 06JPC*
Quartz, feldspar including plagioclase and K-feldspar, illite, chlorite, kaolinite and
dolomite were detected in the study samples. Plagioclase comprises a variety of
anorthite to albite. Microscopic observations of smear slides for the study samples
revealed that quartz and feldspar are the two major minerals in the composition of
detrital grains.
The variation patterns of the Q/F, C/I, CK/I and K/I ratios are different between
fine-grained and glaciomarine units in cores 05JPC/TC and 06JPC (Fig. 3;
Supplementary tables 2–4). The ratios of fine-grained unit are relatively stable
compared with those in glaciomarine units. The higher Q/F ratio in glaciomarine units is
consistent with the finding of previous studies that quartz grains are abundant in the
western Arctic sediments delivered from the Laurentide ice sheet during glacial and
deglacial periods (Bischof et al., 1996; Bischof and Darby, 1997; Phillips and Grantz,
2001; Kobayashi et al., 2016). Some peaks correspond to dolomite-rich layers ("D" in
Fig. 3). Variation in the K/I ratio was associated with that in the Q/F ratio (Fig. 3),
which is in harmony with an idea that kaolinite was delivered via the Beaufort Gyre
circulation (Kobayashi et al., 2016). The C/I and CK/I ratios are lower in glaciomarine
unit than in fine-grained unit in 06JPC (Fig. 3C), which is consistent with the closure of
Bering Strait in the last glacial (Elias et al., 1992), but this difference is not significant
in 05JPC (Fig. 3B).



The Q/F ratio in cores 01A-GC, 05JPC/TC and 06JPC shows a gradual long-term
decrease throughout the Holocene (Fig. 4A). In cores 01A-GC and 06JPC studied in
more detail, the Q/F ratio also indicates millennial- to century-scale variability (Fig. 4A).
Variations of the 5-point running average highlight millennial-scale patterns (Fig. 4A).
The variations are generally asynchronous between both cores on this timescale, which
strongly depends on their age-depth models.
In core 01A-GC, the CK/I and C/I ratios show a general increase after ca. 9.5 ka with
the highest values occurring between 6 and 4 ka, and high ratios around 2.5 ka and 1 ka
(Fig. 4B). In core 06JPC, the ratios show a general increase after 9.2 ka with higher
values occurring between 6 and 3 ka (Fig. 4B). In core 05JPC/TC, slightly higher ratios
occur between 6 and 3 ka after a gradual increase from 9.3 ka (Fig. 4B).

**5. Discussion**
***5.1. Holocene trend in the Beaufort Gyre circulation***
The zonal gradient of the Q/F ratio in western Arctic sediments shown in Fig. 2
suggests that quartz-rich but feldspar-poor sediments are derived from the North
American margin by the BG circulation, whereas feldspar–rich sediments are delivered
to the Chukchi Sea from the Siberian margin by currents along the East Siberian slope,
and the ratio can be used as an index for the BG circulation reflecting changes in its
intensity in sediment-core records (Kobayashi et al., 2016). A consistent upward
decrease in the Q/F ratio in three different cores under study (Fig. 4A) suggests that the
BG weakened during the Holocene. This pattern is consistent with an orbitally-forced
decrease in summer insolation at northern high latitudes from the early Holocene to
present. High summer insolation likely melted sea ice in the Canada Basin, in particular



in the coastal areas (Fig. 5). The evidence of lower ice concentrations at the Canada
Basin margins in the early Holocene was shown in the fossil records of bowhead whale
bones from the Beaufort Sea coast (Dyke and Savelle, 2001) and driftwood from
northern Greenland (Funder et al., 2011). This condition could decrease the stability of
the ice cover at the margins of the Canada Basin, which accelerated the rotation of the
BG circulation (Fig. 5), by comparison with observations from recent decades (Shimada
et al., 2006). A decrease in summer insolation during the Holocene should have
increased the stability of sea-ice cover along the coasts, resulting in the weakening of
the BG.
Recent observations show that the BG circulation is linked to the AO (Proshutinsky
and Johnson, 1997; Rigor et al., 2002). In the negative phase of the AO, the Beaufort
High strengthens and intensifies the BG. If the gradual weakening of the BG during the
Holocene were attributed to atmospheric circulation only, a concurrent shift in the mean
state of the AO from the negative to positive phase would be expected. This view,
however, contradicts the existing reconstructions of the AO history showing multiple
shifts between the positive and negative phase during the Holocene (e.g., Rimbu et al.,
2003; Olsen et al., 2012). We, thus, infer that the decreasing Holocene trend of the BG
circulation is attributed not to changes in the AO pattern, but rather to the increasing
stability of the sea-ice cover in the Canada Basin.
Based on a Holocene sediment record off northeastern Chukchi margin, Darby et al.
(2012) suggested strong positive AO-like conditions between 3 and 1.2 ka based on
abundant ice-rafted iron oxide grains from the West Siberian shelf. In contrast, a mostly
negative AO in the late Holocene can be inferred from mineralogical proxy data
indicating a general decline of the BSI after 4 ka (Ortiz et al., 2009), which could be



attributed to a stronger Aleutian Low (Danielson et al., 2014) that typically corresponds
to the negative AO (Overland et al., 1999). Olsen et al. (2012) also concluded that the
AO tended to be mostly negative from 4.2 to 2.0 ka based on a redox proxy record from
a Greenland lake. In order to comprehend these patterns, we need to consider not only
the atmospheric circulation, but also sea-ice conditions. Based on the Q/F record in this
study, summer Arctic sea-ice cover shrank in the early to middle Holocene, so that fast
ice containing West Siberian grains could less effectively reach the Canada Basin
because sea ice would have melted on the way to the BG (Fig. 5). Later in the Holocene
the ice cover expanded, and West Siberian fast ice could survive and be incorporated
into the BG (Fig. 5). We infer, therefore, that sediment transportation in the BG is
principally governed by the distribution of summer sea ice and the resultant stability of
the ice cover in the Canada Basin.

*5.2. Millennial variability in the BG circulation*

In addition to the decreasing long-term trend, the Q/F ratio in 01A-GC and 06JPC

clearly displays millennial- to century-scale variability (Fig. 4A). Variation in the Q/F
ratio of both 01A-GC and 06JPC indicates a significant periodicity of ~2100 and ~1000
years with weak periodicities of ~500 and ~360 years, consistent with prominent
periodicities in the variation of total solar irradiance (Fig. 6) (Steinhilber et al., 2009). A
comparison with the record of total solar irradiance (Steinhilber et al., 2009) shows a
general correspondence, where stronger BG circulation (higher Q/F ratio) corresponds
to higher solar irradiance (Fig. 7). A ~200-year phase lag between the solar irradiance
and the Q/F ratio in 01A-GC and 06JPC may be attributed to the underestimation of
local carbon reservoir effect. This pattern suggests that millennial-scale variability in





the BG was principally forced by changes in solar irradiance. Because these changes are
energetically much smaller than changes in the summer insolation caused by orbital
forcing, we suppose that solar activity did not directly affect the stability of ice cover in
the Canada Basin. Alternatively, we suggest that the solar activity signal was amplified
by positive feedback mechanisms, possibly through changes in the stability of sea-ice
cover and/or the atmospheric circulation in the northern high latitudes.
In addition to cycles consistent with the solar forcing, Darby et al. (2012) reported a
1,550 year cycle in the Siberian grain variation in the Chukchi Sea record. This cycle
was, however, not detected in our data indicative of the BG variation (Fig. 6). This
difference suggests that the occurrence of Siberian grains in the Chukchi Sea sediments
primarily reflects the formation and transportation of fast ice in the eastern Arctic Ocean
rather than changes in the BG circulation.

*5.3. Holocene changes in the Bering Strait Inflow*
Northward decreasing trends in the CK/I and C/I ratios in surface sediments in the
Chukchi Sea suggests that chlorite-rich sediments are derived from the northern Bering
Sea via Bering Strait, and the ratios can be used as an index for the BSI reflecting
changes in its intensity in sediment-core records (Kobayashi et al., 2016). Although the
variations of the CK/I and C/I ratios are not identical among three study cores (Fig. 4B),
there is a common long-term trend showing a gradual increase from 9 to 4.5 ka and a
decrease afterwards (Fig. 4B). Large fluctuation is significant in 01A-GC from 6 to 4 ka,
and this fluctuation is also seen in 6JPC to some extent (Fig. 4B).
The higher CK/I and C/I ratios in core 01A-GC in the middle Holocene correspond to
higher linear sedimentation rates estimated by interpolation between $^{14}$C dating points,



but this correspondence is not seen in cores 05JPC/TC and 06JPC (Fig. 4C). We assume
that these higher sedimentation rates at 01A-GC indicate intensified BSI, because fine
sediment in the study area is mostly transported by currents from the Bering Sea and
shallow southern Chukchi shelf (Kalinenko, 2001; Darby et al., 2009; Kobayashi et al.,
2016). The difference of chlorite and sedimentation rate records between 01A-GC and
05JPC/06JPC may be related to either 1) variable sediment focusing at different water
depths, or 2) redistribution of the BSI water between different branches after passing the
Bering Strait. 1) A sediment-trap study demonstrated that shelf-break eddies in winter
are important to carry fine-grained lithogenic material from the Chukchi Shelf to the
slope areas (Watanabe et al., 2014). This redeposition process may have weakened the
BSI signal in slope sediments of 05JPC/06JPC compared with outer shelf sediments of
01A-GC. 2) Both the Alaskan Coastal Current (ACC) and the central current can
transport sediment particles to the 05JPC/TC and 06JPC area (red and yellow arrows,
respectively, in Fig. 1; Winsor and Chapman, 2004; Weingartner et al., 2005). In
comparison, the western branch is more likely to carry sediment particles to the site of
01A-GC (blue arrow in Fig. 1). Redistribution of the BSI water may have caused
different response of BSI signals. Although it is not clear which process made the
difference of BSI signals between 01A-GC and 05JPC/06JPC cores, it is highly possible
that the sedimentation rate and mineral composition of 01A-GC are more sensitive to
changes in BSI intensity than those of two other sites.
Diffuse spectral reflectance in core HLY0501-06JPC indicated that chlorite +
muscovite content is especially high in the middle Holocene between ca. 4 and 6 ka
(Supplementary Fig. S1; Ortiz et al., 2009). However, this pattern was not confirmed by
our XRD analysis, where XRD intensities of chlorite and muscovite (detected as illite in

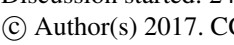



this study) as well as the C/I and CK/I ratios did not show an identifiable enrichment
between 4 and 6 ka (Supplementary Fig. S1). We need more research to understand the
discrepancy of the results.

*5.4. Millennial variability in the BSI*
Variation in the C/I ratio of 01A-GC indicates a significant periodicity of 1900, 1000,
510, 400 and 320 years (Fig. 6A). The 1900, 1000 and 510 years are consistent with
prominent periodicities in the variation of total solar irradiance (Fig. 6C) (Steinhilber et
al., 2009). On the other hands, variation in the C/I ratio of 06JPC indicates a periodicity
of 2200, 830 and 440 years (Fig. 6B). The periodicity is different from that in 01A-GC
(Fig. 6A). This suggests that there are different agents of BSI signals in cores 01A-GC
and 06JPC. In core 01A-GC, 1000-year filtered variation in the C/I ratio is nearly
antiphase with those of the Q/F ratio and total solar irradiance (Steinhilber et al., 2009)
between 0 and 5 ka (Fig. 7). This suggests that millennial-scale variability in the
western branch of the BSI was forced by changes in solar irradiance after 5 ka. Recent
observations demonstrated that the BSI flows northwestward, especially when easterly
winds prevent the ACC (Winsor and Chapman, 2004). Because the easterly winds drive
the BG circulation, this mechanism cannot explain the increase of BSI intensity when
the BG weakened. Alternatively, it is also possible that the solar forcing could
independently regulate the western branch of the BSI via unknown atmospheric-oceanic
dynamics.

*5.5. Ocean circulation, sea ice and biological production*



The BSI, an important carrier of heat to the Arctic, affects sea ice extent in the
Chukchi Sea (e.g., Shimada et al., 2006). Sea ice concentrations in the Chukchi Sea
during the Holocene were reconstructed by dinoflagellate cyst (de Vernal et al., 2005;
2008; 2013; Farmer et al., 2011) and biomarker $IP_{25}$ (Polyak et al., 2016; Stein et al.,

2017).

In central northern Chukchi Sea, $IP_{25}$ records showed that sea ice concentration
indicated by $PIP_{25}$ index in core 01A-GC was lower in 9–7.5 ka and 5.5–4 ka (Fig. 8A;
Stein et al., 2017), suggesting less sea ice conditions in the periods. The low sea ice
concentration during 9–7.5 ka is consistent with the results of previous studies based on
dinoflagellate cyst and $IP_{25}$ records showing the sea ice retreat widely in the Arctic
Ocean, which was attributed to higher summer insolation during the early Holocene
(Dyke and Savelle, 2001; Vare et al., 2009; de Vernal et al., 2013; Stein et al., 2017).
On the other hands, the sea ice retreat during 5.5–4 ka cannot be explained by higher
summer insolation. This period corresponds to that of higher C/I and CK/I ratios
indicative of the stronger BSI at 01A-GC (Fig. 8A). This suggests that the strengthened
BSI during this period contributed to sea ice retreat in the central Chukchi Sea.
In the northeastern Chukchi Sea, dinoflagellate cyst and biomarker $IP_{25}$ records from
several cores in the northeastern Chukchi Sea, including 05JPC, demonstrate that sea
ice concentration in this area was overall higher in the early Holocene than in the
middle and late Holocene (Fig. 8; de Vernal et al., 2005; 2008; 2013; Farmer et al.,
2011; Polyak et al., 2016). This pattern appears to contrast reconstructions from other
Arctic regions that show lower sea-ice concentrations in the early Holocene (de Vernal
et al., 2013). This discrepancy suggests that the intensified BG circulation exported
more ice from the Beaufort Sea to the northeastern Chukchi Sea margin. Furthermore,



502 the heat transport from the North Pacific to the Arctic Ocean by the BSI was likely

503 weaker in the early Holocene than at later times as indicated by the C/I and CK/I ratios

504 of cores 06JPC and 01A-GC (Fig. 8). We infer that this combination of stronger BG

505 circulation and weaker BSI in the early Holocene resulted in increased sea-ice

506 concentration in the northeastern Chukchi Sea despite high insolation levels (Fig. 5). In

507 comparison, intense BSI, a crucial agent of heat transport from the North Pacific to the

508 Arctic Ocean, along with weaker BG in the middle Holocene likely reduced sea ice

509 cover in the Chukchi Sea. During the late Holocene, characterized by the weakest BG

510 and moderate BSI, sea-ice concentrations were intermediate and strongly variable (Fig.

511 8; de Vernal et al., 2008, 2013; Polyak et al., 2016).

512  The nutrient supply by the BSI potentially affects marine production in the Chukchi

513 Sea. We tested this possibility to compare our BSI record with marine production

514 records from cores 01A-GC (Park et al., 2016; Stein et al., 2017). Isoprenoid GDGTs

515 and brassicaterol showed concentration maxima during the periods between 8 and 7.5

516 ka and 6 and 4.5 ka (Fig. 8A). Isoprenoid GDGTs are produced by marine Archaea

517 (Nishihara et al., 1987) that use ammonia, urea and organic matter in the water column

518 (Qin et al., 2014). Brassicasterol is known as a sterol which is abundant in diatoms

519 (Volkman et al., 1986). Their abundance can, thus, be used as proxies to indicate marine

520 production in the water column. The periods with abundant isoprenoid GDGTs and

521 brassicasterol corresponded to the periods of low $PIP_{25}$ indicative of less sea ice (Fig.

522 8A). This correspondence suggests that the biological productivity increased with the

523 retreat of sea ice in the Chukchi Sea during the middle Holocene. The BSI indices, the

524 C/I and CK/I ratios, showed a maximum between 6 and 4 ka, which corresponded to the

525 periods of high marine production, but the corresponding maximum between 8 and 6.5



ka is not significant. Also, correspondence between the BSI indices and biomarker
concentrations are not clear after 4 ka. This suggests that marine production was not a
simple response to nutrient supply but was affected by other processes such as the
increase of irradiance in the water column (Frey et al., 2011; Lee and Whitledge, 2005)
and wind-induced mixing that replenishes sea surface nutrients (Carmack et al., 2006).

*5.6. Causes of BSI variations*

Chukchi Sea sedimentary core records indicate a considerable variability in the BSI

intensity, with a common long-term trend of a gradual increase from 9 to 4.5 ka and a
decrease afterwards (Fig. 4B). Below we discuss the possible controls on this
variability.

The timing of the initial postglacial flooding of the ~50-m-deep Bering Strait was

estimated as between ca. 12 and 11 ka (Elias et al., 1992; Keigwin et al., 2006). Gradual
intensification of the BSI inferred from the increase in chlorite content from ca. 9 to 6
ka may have been largely controlled by the widening and deepening of the Bering Strait
with rising sea level, although other factors as discussed below yet need to be tested.
After the sea level rose to nearly present position by ca. 6 ka, its influence on changes in
the BSI volume was negligible.

The possible driving forces of the BSI at full interglacial sea level may include

several controls. One is related to the sea surface height difference between the Pacific
and Atlantic Oceans regulated by the atmospheric moisture transport from the Atlantic
to the Pacific Ocean across Central America (Stigebrandt, 1984). Increase in this
moisture transport during warm climatic intervals (Leduc et al., 2007; Richter and Xie,
2010; Singh et al., 2016) may have intensified the BSI. Salinity proxy data for the last



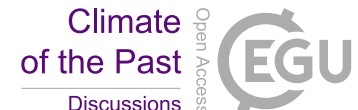

90 ka from the Equatorial East Pacific confirm increased precipitation during warm
events, but also show the trans-Central America moisture transport may operate
efficiently only during intervals with a northerly position of the Intertropical
Convergence Zone due to orographic constraints (Leduc et al., 2007). The existing
Holocene salinity records from the North Pacific (e.g., Sarnthein et al., 2004) do not yet
provide sufficient material to test the impact of these changes on the BSI.

Interplay of the global wind field and the AMOC has been proposed as another

potential control on the BSI (De Boer and Nof, 2004; Ortiz et al., 2012). Results of an
analytical ocean modeling experiment (Sandal and Nof, 2008) based on the island rule
(Godfrey, 1989) suggest that weaker Subantarctic Westerlies in the middle Holocene
could decrease the near surface, cross-equatorial flow from the Southern Ocean to the
North Atlantic, thus enhancing the BSI and Arctic outflow into the Atlantic. This
hypothesis waits to be tested more thoroughly, including robust proxy records of the
Subantarctic Westerlies over the Southern Ocean.

Finally, BSI can be controlled by the regional wind patterns in the Bering Sea

(Danielson et al., 2014), as explained above in Section 2.1. Oceanographic observations
of 2000–2011 clearly show a decadal response of the BSI to a change in the sea level
pressure in the Aleutian Basin affecting the dynamic sea surface height along the Bering
Strait pressure gradient. In order to conclude, if this relationship holds on longer time
scales, longer-term records are needed from areas affected by the BSI and the Bering
Sea pressure system.

A number of proxy records from the Bering Sea and adjacent regions, both marine

and terrestrial, have been used to characterize paleoclimatic conditions related to
changes in the Bering Sea pressure system (e.g., Barron et al., 2003; Anderson et al.,




2005; Katsuki et al., 2009; Barron and Anderson, 2011; Osterberg et al., 2014). Various
proxies used in these records consistently show that the Aleutian Low was overall
weaker in the middle Holocene than in the late Holocene, opposite to the BSI strength
inferred from our Chukchi Sea data (Fig. 4B). For example, multi-proxy data from the
interior Alaska and adjacent territories (Kaufman et al., 2016, and references therein)
indicate overall drier and warmer conditions in the middle Holocene, consistent with
weaker Aleutian Low and stronger BSI. Diatom records from southern Bering Sea
indicate more abundant sea ice in the middle Holocene, also suggestive of a weaker
Aleutian Low (Katsuki et al., 2009). Alkenone and diatom records from the California
margin show that the sea surface temperature was lower in the middle Holocene,
suggesting stronger northerly winds indicative of weaker Aleutian Low (Barron et al.,
2003). Intensification of the Aleutian Low in the late Holocene, which follows from
these results, would have decreased sea level pressure in the Aleutian Basin, and thus
the strength of the BSI, consistent with overall lower BSI after ca. 4 ka inferred from
the Chukchi Sea sediment-core data (Fig. 4). A considerable climate variability of the
Bering Sea region captured in the upper Holocene records, some of which have very
high temporal resolution, is also closely linked to the pressure system changes
(Anderson et al., 2005; Porter, 2013; Osterberg et al., 2014; Steinman et al., 2014). In
particular, weakening of the Aleutian Low is reflected in Alaskan ice (Porter, 2013;
Osterberg et al., 2014) and lake cores (Anderson et al., 2005; Steinman et al., 2014) at
intervals centered around ca. 2 and 1–0.5 ka BP, which may correspond to BSI
increases in the Chukchi core 01A-GC at ca. 2.5 and 1 ka BP (Fig. 4), considering the
uncertainties of the sparse age constraints in the upper Holocene and/or underestimation
of reservoir ages. Overall, the Aleutian Low control on the BSI on century to millennial



time scales is corroborated by ample proxy data in comparison with the other potential
controls, although more evidence is still required for a comprehensive interpretation.

***6. Conclusions***

The sedimentary proxy-based reconstruction of the BG weakening during the

Holocene, likely driven by the orbitally-controlled summer insolation decrease,
indicates basin-wide changes in the Arctic current system and suggests that the stability
of sea ice is a key factor regulating the Arctic Ocean circulation on the long-term (e.g.,
millennial) time scales. This conclusion helps to better understand a dramatic change in
the BG circulation during the last decade, probably caused by sea-ice retreat along the
margin of the Canada Basin and a more efficient transfer of the wind momentum to the
ice and underlying waters (Shimada et al., 2006). These results suggest that the rotation
of the BG is likely to be further accelerated by the projected future retreat of summer
Arctic sea ice. Millennial to multi-centennial variability in the quartz/feldspar ratio (the
BG circulation) is consistent with fluctuations in solar irradiance, suggesting that solar
activity affected the BG strength on these timescales.

Our results on clay-mineral ratios quantifying inputs of chlorite from the Bering Sea

to sediments at the northern Chukchi margin provide a robust record of the strength of
the BSI during the Holocene. We conclude that BSI variability after the establishment
of the full interglacial sea level was primarily controlled by the Bering Sea pressure
system (strength and position of the Aleutian Low). Details of this mechanism, as well
as contributions from other potential BSI controls, such as climatically-driven
Atlantic-Pacific moisture transfer and the impact of global wind stress, need to be
further investigated.




**Acknowledgements**

We thank all of the captain, crew and scientists of RV *Araon* for their help during the cruise of sampling. We also thank Yu-Hyeon Park, Anne de Vernal, Seth L. Danielson, Julie Brigham-Grette and Kaustubh Thirumalai for valuable discussion, So-Young Kim, Hyo-Sun Ji, Young-Ju Son, Duk-Ki Han and Hyoung-Jun Kim for assistance in coring and subsampling and Keiko Ohnishi for analytical assistance. The study was supported by a grant-in-aid for Scientific Research (B) the Japan Society for the Promotion of Science, No. 25287136 (to M.Y.) and Basic Research Project (PE16062) of Korean Polar Research Institute and the NRF of Korea Grant funded by the Korean Government (NRF-2015M1A5A1037243) (to S.I.N.).

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





**Figure captions**

Fig. 1. Index map showing location of cores ARA02B 01A-GC (this study),

HLY0501-05JPC/TC (this study and Farmer et al., 2011), HLY0501-06JPC (this study

and Ortiz et al., 2009), and HLY0205-GGC19 (Farmer et al., 2011), as well as surface

sediment samples (Kobayashi et al., 2016, with additions). BSI = Bering Strait inflow,

BC = Barrow Canyon, HN = Hanna Shoal, and HR = Herald Shoal. BG = Beaufort

Gyre, ACC = Alaskan Coastal Current, SBC = Subsurface Boundary Current, ESCC =

East Siberian Coastal Current, TPD = Transpolar Drift. Red, yellow and blue arrows

indicate BSI branches. AO+ and AO– indicate circulation in the positive and negative

phases of the Arctic Oscillation, respectively.

Fig. 2. Spatial distributions of the diffraction intensity ratios of (A) feldspar to quartz

(Q/F), and of (B) chlorite+kaolinite and (C) chlorite to illite (CK/I and C/I, respectively)

of bulk sediments, and (D) the longitudinal distribution of the Q/F ratio in the western

Arctic (>65°N) and (E) the latitudinal distribution of the CK/I and C/I ratios in the

Bering Sea and the western Arctic (>150°W). The C/I ratio could not be determined in

some coarse-grained sediment samples. Data from Kobayashi et al. (2016) with

additions for the Beaufort Sea (See supplementary Table 1 in more detail).

Fig. 3. Depth profile in (A) quartz/feldspar (Q/F) ratio, (chlorite + kaolinite)/illite

(CK/I), chlorite/illite (C/I) and kaolinite/illite (K/I) ratios with 1σ-intervals (analytical

error) in cores (A) ARA02B 01A-GC, (B) HLY0501-05JPC/TC and (C)




HLY0501-06JPC (Supplementary Tables 2–4). "D" indicates a dolomite-rich layer.
Note that the depth scale of 01A-GC is doubled.

Fig. 4. Changes in (A) quartz/feldspar (Q/F) ratio and the June insolation at 75°N, (B)
(chlorite + kaolinite)/illite (CK/I) and chlorite/illite (C/I) ratios and (C) linear
sedimentation rates (LSR) in cores ARA02B 01A-GC, HLY0501-05JPC/TC and
HLY0501-06JPC during the last ca. 9.3 ka.

Fig. 5. Conceputual map showing the distribution of summer sea ice and the rotation of
the Beaufort Gyre (BG) in the early, middle and late Holocene, inferred from the
quartz/feldspar (Q/F) proxy record. Also shown is the Bering Strait inflow (BSI)
intensity inferred from the (chlorite + kaolinite)/illite (CK/I) and chlorite/illite (C/I)
ratios. Red arrow indicates the drift path of Kara Sea grains (KSG; Darby et al., 2012).

Fig. 6. Max Entropy power spectra of variation in the quartz/feldspar (Q/F) and
chlorite/illite (C/I) ratios in core ARA02B 01A-GC (N=85, m=21) and
HYL0501-06JPC (N=79, m=22) during 1.4–7.9 ka and the total solar irradiance (N=932,
m=140)(Steinhilber et al., 2009) during the last 9.3 ka.

Fig. 7. Detrended variations in the solar irradiance (TSI; Steinhilber et al., 2009), the
quartz/feldspar (Q/F) ratio in logarithmic scale in cores ARA02B 01A-GC and
HYL0501-06JPC and the chlorite/illite (C/I) ratio in core ARA02B 01A-GC during the
Holocene, with 400-year moving averages and 1,000-year filtered variations indicated




by dark colored and black lines, respectively. The detrended values were obtained by
cubic polynomial regression.

Fig. 8. Changes in (A) (chlorite + kaolinite)/illite (CK/I) and chlorite/illite (C/I) ratios,
PIP$_{25}$ (P$_D$IP$_{25}$ and P$_B$IP$_{25}$ based on IP$_{25}$ and dinosterol or brassicasterol concentrations)
indices (Stein et al., 2017), and isoprenoid GDGT (Park et al., 2016) and brassicasterol
concentrations (Stein et al., 2017) in core ARA02B 01A-GC, (B) CK/I and C/I ratios in
core HLY0510-5JPC/TC, IP$_{25}$ concentrations in core HLY0510-5JPC (Polyak et al.,
2016), mean annual sea ice cover concentration (scale from 0 to 10) estimated from
dinoflagellate cyst assemblages in cores 05JPC and GGC19 (Farmer et al., 2011; de
Vernal et al., 2013).





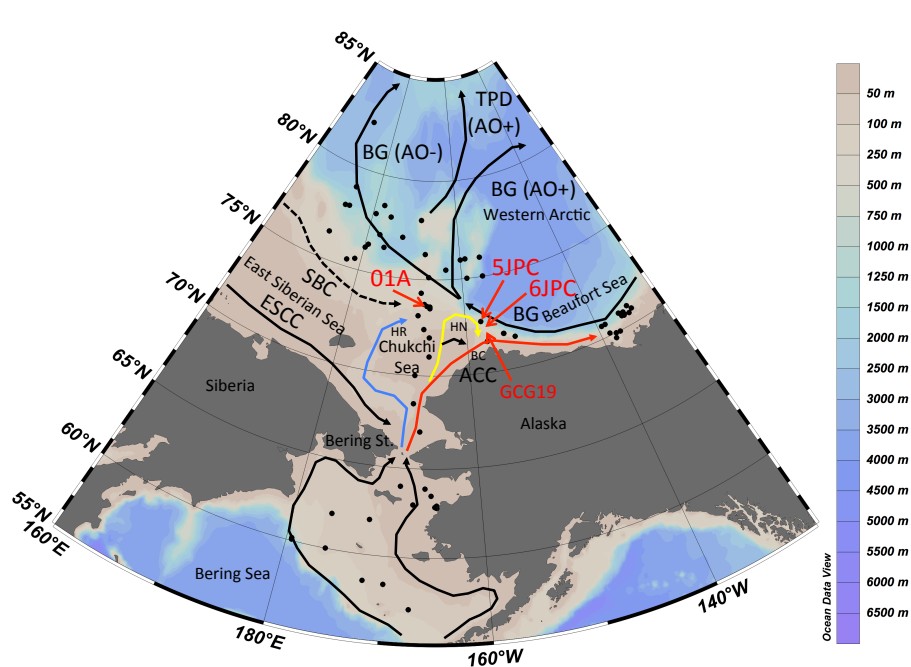



Fig. 1

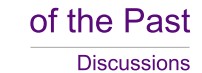



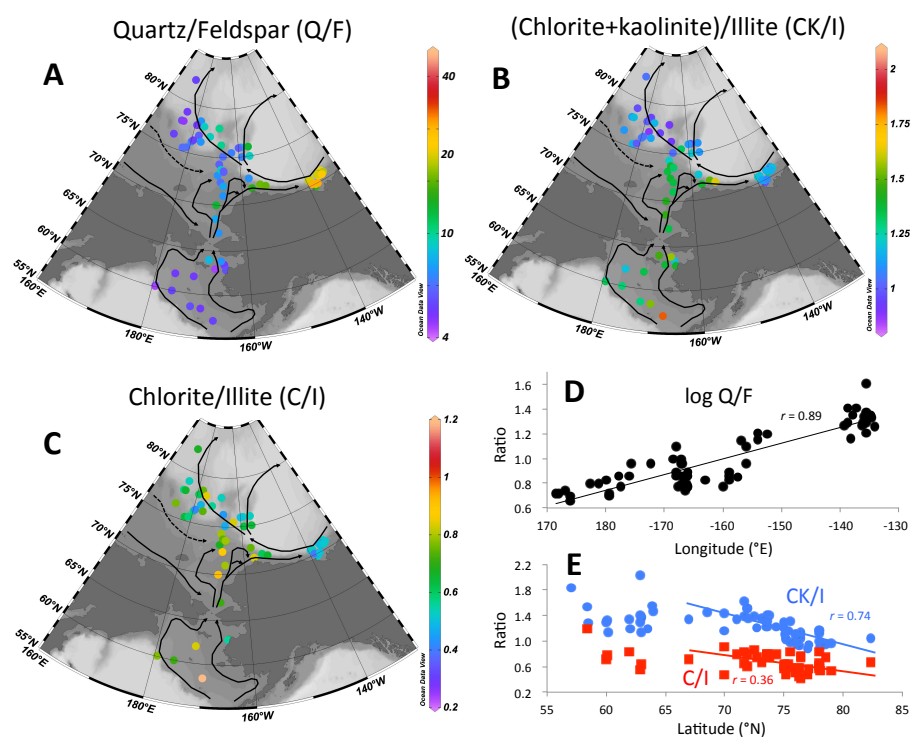


Fig. 2





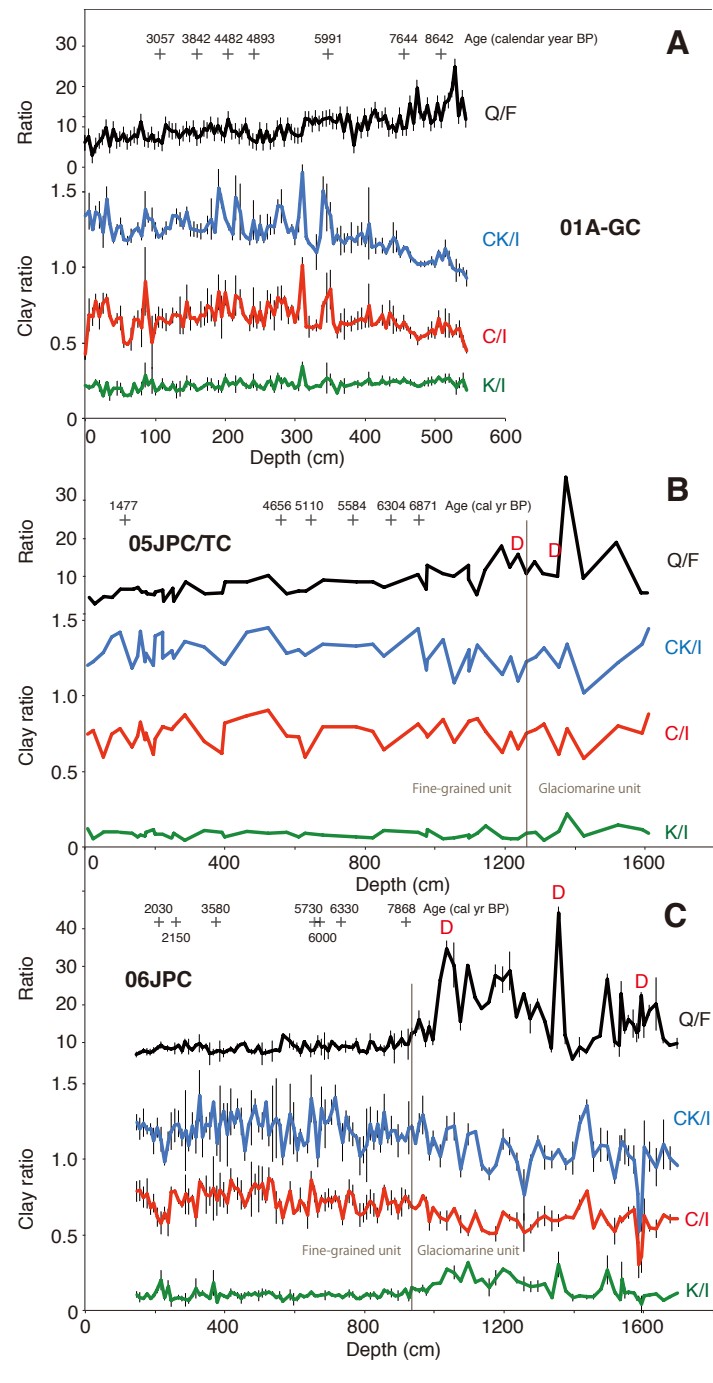


Fig. 3





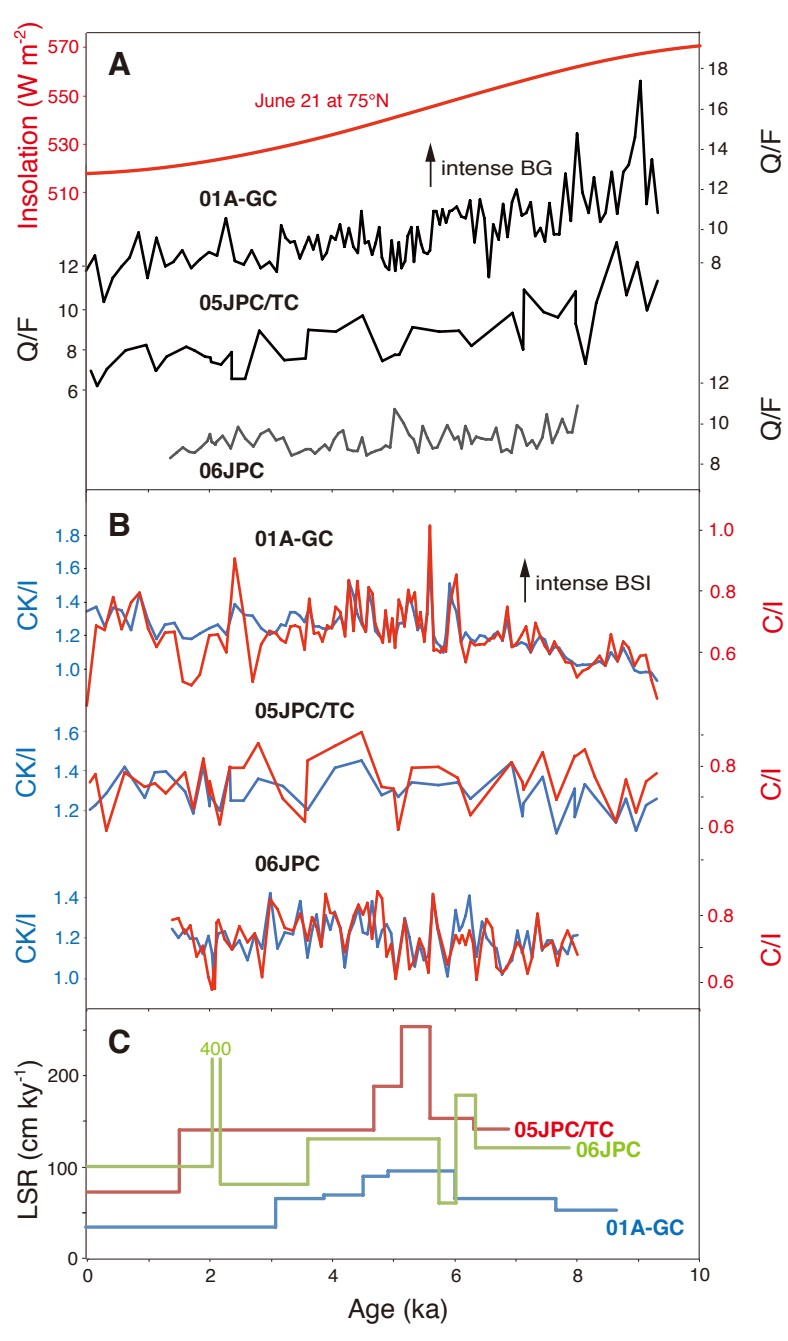


Fig. 4.





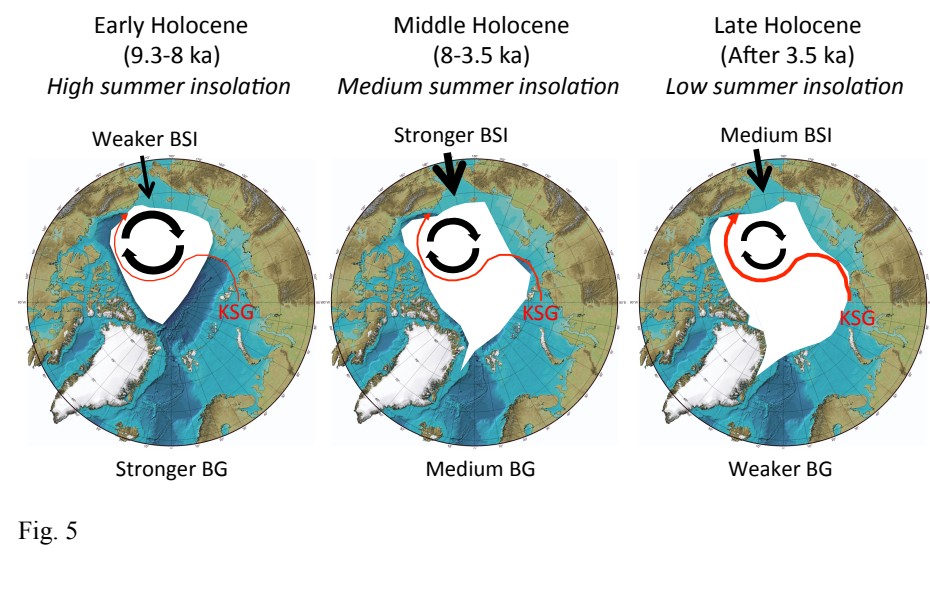


Fig. 5




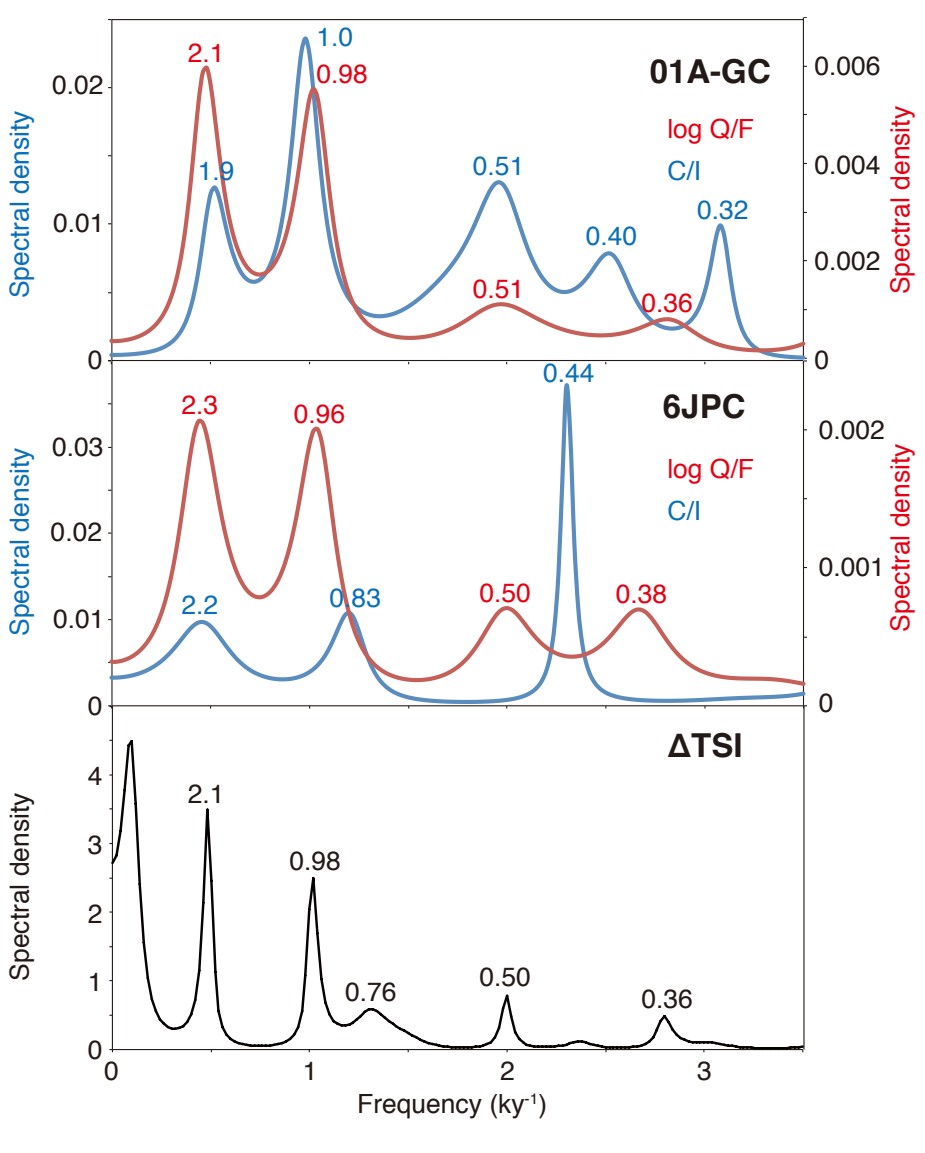


Fig. 6





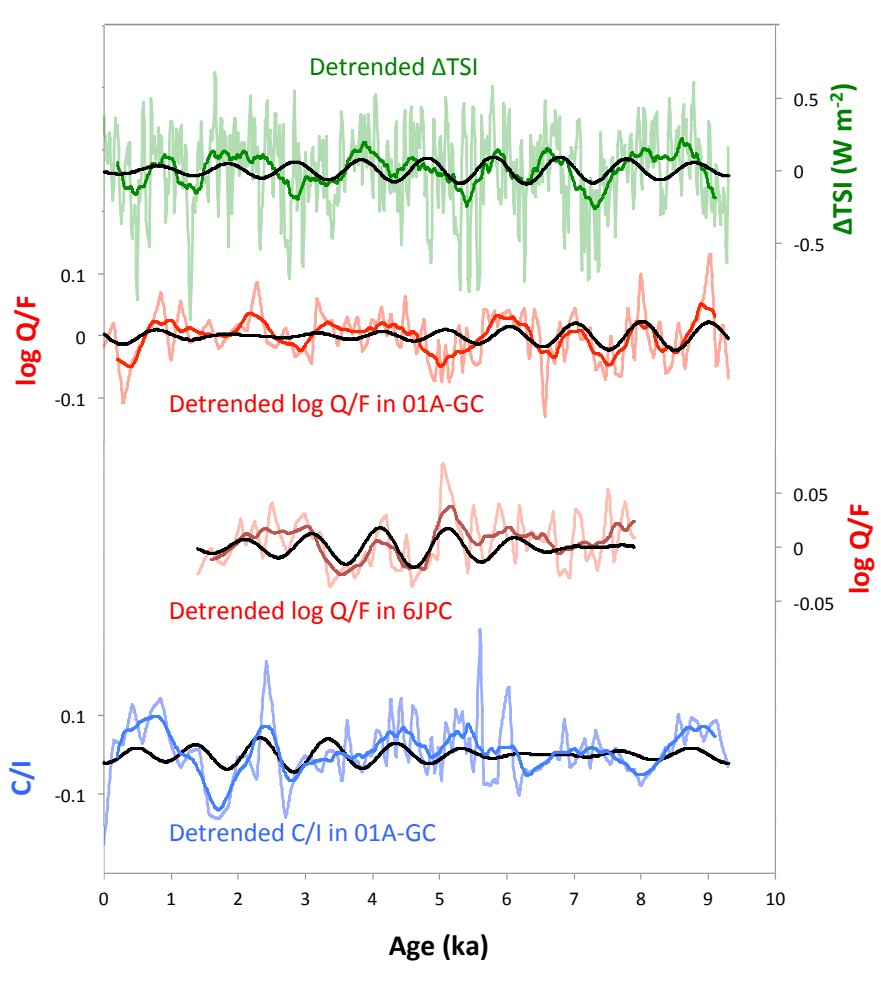


Fig. 7



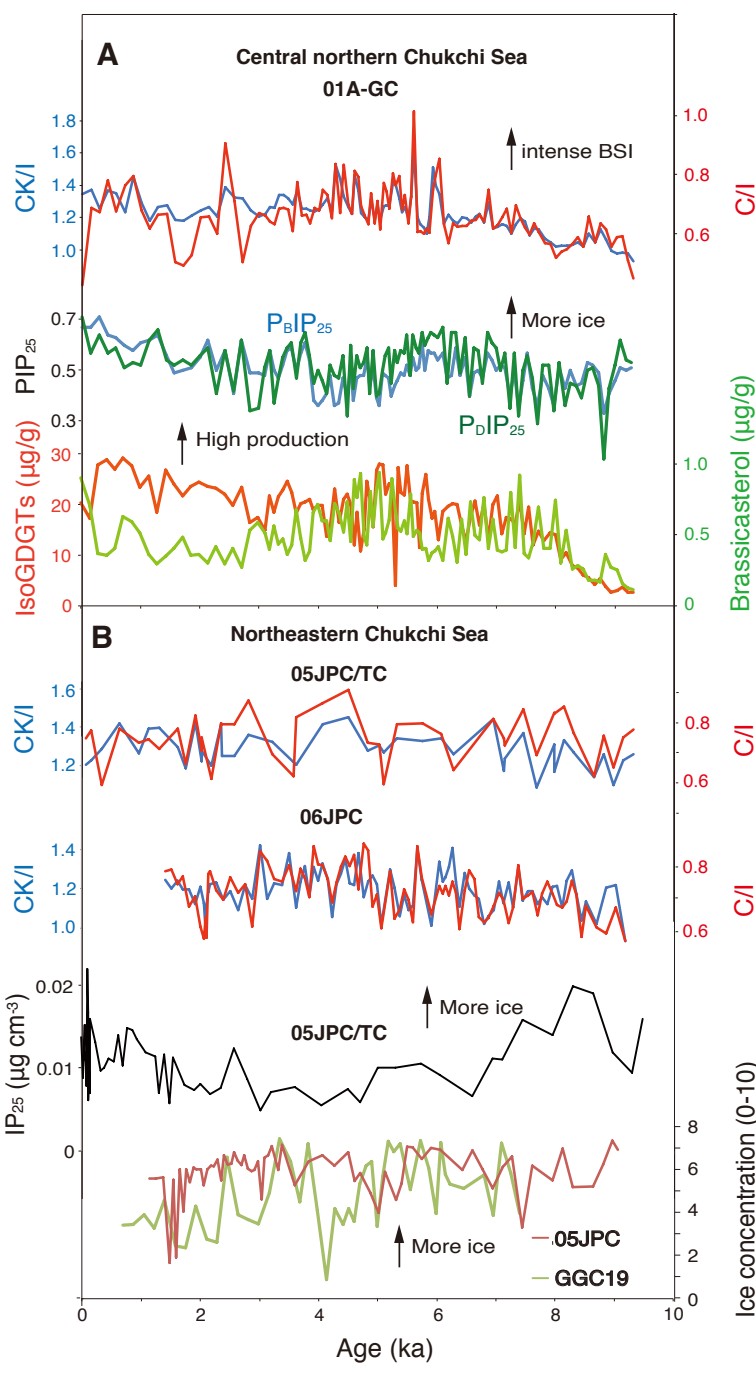


Fig. 8