# Peer review of "Holocene dynamics in the Bering Strait inflow to the Arctic and the Beaufort Gyre"

_Climate of the Past, 2017_

## Short Comment (SC1)

[revised manuscript text omitted]

Fig. 1

[Figure]

[Figure]

[Figure]

Fig. 2

[Figure]

[Figure]

Fig. 3

[Figure]

[Figure]

[Figure]

Fig. 4.

[Figure]

[Figure]

[Figure]

Fig. 5

[Figure]

[Figure]

[Figure]

Fig. 6

[Figure]

[Figure]

[Figure]

Fig. 7

[Figure]

[Figure]

Fig. 8

---

## Referee Comment (RC1) · Anonymous Referee #1 · 9 Jun 2017

This paper deals with sediment cores from the Chukchi Sea and uses XRD mineralogy to study variability of the Beaufort Gyre and Pacific inflow into the Arctic Ocean during the Holocene.

This submission is a revised version of an earlier manuscript published in Climate of the Past Discussions.

One of the main comments on the original manuscript was the over-interpretation of results and linkage to Atlantic teleconnections. This component is toned down here,

which has improved the manuscript.

Several other reviewers' comments from the original remain, however, unaddressed so some are repeated here.

This study provides a wealth of new data and new insights on the Chukchi Sea in the Holocene. I can recommend publication of this manuscript, provided the authors address the following comments and suggestions for revision.

Problems with C/I and (C+K)/I as proxies for Bering Strait inflow: - how solid is this proxy, if it does not show any difference (in core 5JPC, Figure 3B) between the Holocene and the last glacial when the strait was closed? - The records from the three cores show very little agreement for these proxies. Again, what does this mean for the proxy? It does not seem a convincing record of Bering inflow.

Page 9. Lines 206-210. The top of core 01A-GC is assumed to be of modern age, because the authors write that sterols and IP25 show a decreasing trend in the top 10 cm (Stein et al 2017). This is a very poor indicator of recovery of the top sediments. Looking at the data in Stein et al 2017, the statement is not even accurate. The variability in the top 10 cm is of the same order of magnitude as deeper in the core. I suggest that this is removed (lines 206-210) and that it is acknowledged that the core top age is uncertain. There are no Pb210 dates, or a surface core to correlate with.

There should be a table with radiocarbon dates and paleointensity datums (depth, age, reference). It would summarize the information spread out over pages 9-10 and shown in Figure 3. I suggest bringing back Table 1 from the original submission, adding the magnetic datums, and addressing the original reviewer comments to this version.

Divide section 3 in subsections: e.g. 3.1 Coring and Sampling, 3.2 Chronology, 3.3 XRD Mineralogy

Figure 2 - From Panel E, one can see that there should be a data point with a CK/I ratio around 2.0 at about 63°N. This is not visible in Panel B. Check this carefully, as there

may be others? - At some sites, there are too many data points for this type of plot. An example: In Panel A, at the Mackenzie delta there are a lot of yellow dots, but they are covering up green ones as well. Either, make inserts for those areas, or make the dots smaller? - Panel E. The regression lines in CK/I and C/I vs latitude do not extend further south than 65N. Correct this or explain why.

Figure 3. What do the crosses represent? Radiocarbon dates, paleointensity datums? Please specify. Add them all to a table (perhaps supplementary).

Figure 3. Rather than showing "D" for dolomite rich layers, please show the actual dolomite data. Also, add to the methods how dolomite was quantified (lines 250-260), and add the data to the supplementary tables.

Figure 3B. Please make it possible to distinguish between samples from the piston core vs trigger core by using different symbols.

Figure 4B. Same comment. Around 4000 cal yrs BP, there seem to be two data points for the same age. Is one JPC and one TC? The difference in their C/I values are large. Does this illustrate the uncertainty of the method?

Page 22 line 515. Correct "brassicasterol".

Page 23 line 538. Add citation to Jakobsson et al 2017 Climate of the Past (this same special issue).

---

## Author Comment (AC1) · 19 Jun 2017

Reply to the interactive comment of anonymous referee #1 on "Holocene dynamics in the Bering Strait inflow to the Arctic and the Beaufort Gyre circulation based on sedimentary records from the Chukchi Sea" by Masanobu Yamamoto et al.

We thank anonymous referee #1 for his/her helpful comments on our manuscript. Below is our reply to the main comments.

Comment: This paper deals with sediment cores from the Chukchi Sea and uses XRD

[Figure]

mineralogy to study variability of the Beaufort Gyre and Pacific inflow into the Arctic Ocean during the Holocene. This submission is a revised version of an earlier manuscript published in Climate of the Past Discussions. One of the main comments on the original manuscript was the over-interpretation of results and linkage to Atlantic teleconnections. This component is toned down here, which has improved the manuscript. Several other reviewers' comments from the original remain, however, unaddressed so some are repeated here. This study provides a wealth of new data and new insights on the Chukchi Sea in the Holocene. I can recommend publication of this manuscript, provided the authors address the following comments and suggestions for revision.

Reply: Thank you for recognizing the significance of our paper. We will revise it according to your suggestions.

Comment: Problems with C/I and (C+K)/I as proxies for Bering Strait inflow: - how solid is this proxy, if it does not show any difference (in core 5JPC, Figure 3B) between the Holocene and the last glacial when the strait was closed? - The records from the three cores show very little agreement for these proxies. Again, what does this mean for the proxy? It does not seem a convincing record of Bering inflow.

Reply: Indeed, two samples near the bottom (1600 cm) of core 5JPC have the same CK/I and C/I ratios as those of Holocene sediments. However, glacial/deglacial depositional and circulation environments were very different from the Holocene, as exemplified by abundant detrital carbonates with the Laurentide provenance. Likewise, under environments non-analogous to the Holocene, clay minerals may have had a different provenance, with chlorite possibly transported from a source other than the Bering Sea. Some intervals in the deglacial unit in 05JPC are characterized by high abundance of kaolinite and terrestrial soil organic matter (branched GDGTs), probably delivered from inland North America by deglacial discharge (Suzuki et al., AGU fall meeting 2016). Chlorite may have also been delivered from areas affected by the Laurentide glaciation this period. The bottom line is that glacial/deglacial records

cannot be used for characterizing Holocene conditions. In comparison, the spatial distribution of clay minerals in surface sediments suggests that the Bering Strait inflow provides a major contribution of chlorite-rich sediments under modern settings. As depositional conditions in the Chukchi Sea do not appear to have changed principally in the Holocene, there is enough reason to apply the modern-type provenance pattern to understanding Holocene changes in the Bering Strait inflow. We also recognize somewhat different patterns of C/I and CK/I among the three cores investigated. We are assuming that such a difference can be attributed to variable sediment focusing at different water depth and redistribution of the Bering Strait water between different branches after passing Bering Strait (lines 435-450). Further studies using more cores, e.g., from a depth transect, are required to clarify this issue.

Comment: Page 9. Lines 206-210. The top of core 01A-GC is assumed to be of modern age, because the authors write that sterols and IP25 show a decreasing trend in the top 10 cm (Stein et al 2017). This is a very poor indicator of recovery of the top sediments. Looking at the data in Stein et al 2017, the statement is not even accurate. The variability in the top 10 cm is of the same order of magnitude as deeper in the core. I suggest that this is removed (lines 206-210) and that it is acknowledged that the core top age is uncertain. There are no Pb210 dates, or a surface core to correlate with. There should be a table with radiocarbon dates and paleointensity datums (depth, age, reference). It would summarize the information spread out over pages 9-10 and shown in Figure 3. I suggest bringing back Table 1 from the original submission, adding the magnetic datums, and addressing the original reviewer comments to this version.

Reply: We agree that the core top in ARA 01-GC may not represent the modern age due to some sediment loss in the coring process. This is indicated by the absence of oxidized brown sediment at the core top, as opposed to a multi-corer collected at the same site. Nevertheless, we believe that the top of 01-GC is close to the sediment surface based on the biomarker distribution. Fig. 1 (below) shows the concentration profile of IP25 and brassicasterol (Stein et al., 2017). We suppose that the downward

decrease in concentrations of both compounds in the top 10 cm indicates their degradation with burial. A similar extent of brassicasterol concentration decrease occurs also in some of the deeper intervals, but is unique for the upper ∼200 cm, while the IP25 decrease at the top is unique for the entire record. We will provide according explanations to this part and indicate that the core-top age is uncertain. We will also bring back Table 1 with the paleomagnetic datums.

Comment: Divide section 3 in subsections: e.g. 3.1 Coring and Sampling, 3.2 Chronology, 3.3 XRD Mineralogy

Reply: We will divide section 3 into subsections as suggested.

Comment: Figure 2 - From Panel E, one can see that there should be a data point with a CK/I ratio around 2.0 at about 63_N. This is not visible in Panel B. Check this carefully, as there may be others? - At some sites, there are too many data points for this type of plot. An example: In Panel A, at the Mackenzie delta there are a lot of yellow dots, but they are covering up green ones as well. Either, make inserts for those areas, or make the dots smaller? - Panel E. The regression lines in CK/I and C/I vs latitude do not extend further south than 65N. Correct this or explain why.

Reply: The symbol of the sample having a CK/I of 2.0 in the Yukon River estuary is hidden by another sample in Fig. 2B. We will make an enlarged map for Yukon and Mackenzie River estuary area and put it in either Fig 2 or supplementary material. The regression lines show the trend for the Chukchi Sea. This suffices to show a northward decrease of the ratios north of Bering Strait. The Bering Sea sediments do not show a systematic trend, probably reflecting multiple sources of chlorite, such as the Yukon River, Aleutian Island, etc. We will add according explanations.

Comment: Figure 3. What do the crosses represent? Radiocarbon dates, paleointensity datums? Please specify. Add them all to a table (perhaps supplementary).

Reply: Crosses represent radiocarbon dates in 01-GC and 5JPC and paleointensity

datums in 06JPC. We will specify that in the caption and add this information in a table.

Comment: Figure 3. Rather than showing "D" for dolomite rich layers, please show the actual dolomite data. Also, add to the methods how dolomite was quantified (lines 250-260), and add the data to the supplementary tables.

Reply: Dolomite intensity will be shown in Figure 3, and the method will be added to the text. The data will be presented in a supplementary table.

Comment: Figure 3B. Please make it possible to distinguish between samples from the piston core vs trigger core by using different symbols.

Reply: We will show different symbols or colors.

Comment: Figure 4B. Same comment. Around 4000 cal yrs BP, there seem to be two data points for the same age. Is one JPC and one TC? The difference in their C/I values are large. Does this illustrate the uncertainty of the method?

Reply: Both samples were derived from core 5JPC (392 and 398 cm). The difference in the values is larger than the analytical error. We assume that this difference could be related to a high-amplitude fluctuation that was observed at the same stratigraphic level in core 01-GC. We will add an according explanation.

Comment: Page 22 line 515. Correct "brassicasterol".

Reply: This will be corrected.

Comment: Page 23 line 538. Add citation to Jakobsson et al 2017 Climate of the Past (this same special issue).

Reply: Jakobsson et al. (2017) will be cited.

———————————————————

Brassicasterol (µg/gSed)    IP25 (µg/gSed)

**Fig. 1.** Concentrations of IP25 and brassicasterol in sediments from core ARA02B 01A-GC (Stein et al., 2017)

---

## Referee Comment (RC2) · T. M. Cronin (Referee) · 7 Jul 2017

[Figure]

[Figure]

[Figure]

[Figure]

Fig. 1

[Figure]

[Figure]

[Figure]

Fig. 2

[Figure]

[Figure]

Fig. 3

[Figure]

[Figure]

[Figure]

Fig. 4.

[Figure]

[Figure]

[Figure]

Fig. 5

[Figure]

[Figure]

[Figure]

Fig. 6

[Figure]

[Figure]

[Figure]

Fig. 7

[Figure]

[Figure]

Fig. 8

[referee-annotated manuscript omitted]

---

## Author Response (AR1)

July 21, 2017

*Climate of the Past* Editorial Office

Dear Martin,

We would like to submit our revised manuscript entitled "Holocene dynamics in the Bering Strait inflow to the Arctic and the Beaufort Gyre circulation based on sedimentary records from the Chukchi Sea" by Yamamoto et al. We thank you, Dr. Cronin and an anonymous reviewer for helpful comments. We revised the manuscript according to your and reviewers' comments.

Editor's comments: *Your paper has now been seen by two reviewers. Both find the paper worth publication after revision. I concur with the two reviewers as I also find that this version has been considerably improved with respect to the main critique raised by the reviewers on your earlier submitted version; that the data was over interpreted. In your point by point response to reviewer 1, it is explained how the comments will be addressed in a revision. I find that this is all in order, please infer the corrections as you have suggested you will do. The discussion on the significance of the proxy C/I and (C+K)/I, whether or not it is capable of capturing variations of the Bering Strait inflow, is crucial to the conclusions. For this reason, I would like to see some more of the reasoning you make in the interactive comment in the actual paper. I therefore suggest that you add a bit in the Discussion on this topic. I look forward to see your revised version of the paper considering the comments made by the two reviewers.*
Reply: Thank you for your decision and comments. We revised our manuscript according to both reviewers' comments. The reasoning we make in the interactive comment is added in Summary and Conclusions section in lines 733 to 741.

Reply to anonymous referee #1 on "Holocene dynamics in the Bering Strait inflow to the Arctic and the Beaufort Gyre circulation based on sedimentary records from the Chukchi Sea" by Masanobu Yamamoto et al.

We thank anonymous referee #1 for his/her helpful comments on our manuscript. Below is our reply to the main comments.

*Comment: This paper deals with sediment cores from the Chukchi Sea and uses XRD mineralogy to study variability of the Beaufort Gyre and Pacific inflow into the Arctic Ocean during the Holocene. This submission is a revised version of an earlier manuscript published in Climate of the Past Discussions. One of the main comments on the original manuscript was the over-interpretation of results and linkage to Atlantic teleconnections. This component is toned down here, which has improved the manuscript. Several other reviewers' comments from the original remain, however, unaddressed so some are repeated here. This study provides a wealth of new data and new insights on the Chukchi Sea in the Holocene. I can recommend publication of this manuscript, provided the authors address the following comments and suggestions for revision.*

Reply: Thank you for recognizing the significance of our paper. We revised it according to your suggestions.

*Comment: Problems with C/I and (C+K)/I as proxies for Bering Strait inflow: - how solid is this proxy, if it does not show any difference (in core 5JPC, Figure 3B) between the Holocene and the last glacial when the strait was closed? - The records from the three cores show very little agreement for these proxies. Again, what does this mean for the proxy? It does not seem a convincing record of Bering inflow.*

Reply: Indeed, two samples near the bottom (1600 cm) of core 5JPC have the same CK/I and C/I ratios as those of Holocene sediments. However, glacial/deglacial depositional and circulation environments were very different from the Holocene, as exemplified by abundant detrital carbonates with the Laurentide provenance. Likewise, under environments non-analogous to the Holocene, clay minerals may have had a different provenance, with chlorite possibly transported from a source other than the Bering Sea. Some intervals in the deglacial unit in 05JPC are characterized by high abundance of kaolinite and terrestrial soil organic matter (branched GDGTs), probably delivered from inland North America by deglacial discharge (Suzuki et al., AGU fall meeting 2016). Chlorite may have also been delivered from areas affected by the Laurentide glaciation this period.

The bottom line is that glacial/deglacial records cannot be used for characterizing Holocene conditions. In comparison, the spatial distribution of clay minerals in surface sediments suggests that the Bering Strait inflow provides a major contribution of chlorite-rich sediments under modern settings. As depositional conditions in the Chukchi Sea do not appear to have changed principally in the Holocene, there is enough reason to apply the modern-type provenance pattern to understanding Holocene changes in the Bering Strait inflow.

We also recognize somewhat different patterns of C/I and CK/I among the three cores investigated. We are assuming that such a difference can be attributed to variable sediment focusing at different water depth and redistribution of the Bering Strait water between different branches after passing Bering Strait (lines 549 to 564). Further studies using more cores, e.g., from a depth transect, are required to clarify this issue.

*Page 9. Lines 206-210. The top of core 01A-GC is assumed to be of modern age, because the authors write that sterols and IP25 show a decreasing trend in the top 10 cm (Stein et al 2017). This is a very poor indicator of recovery of the top sediments. Looking at the data in Stein et al 2017, the statement is not even accurate. The variability in the top 10 cm is of the same order of magnitude as deeper in the core. I suggest that this is removed (lines 206-210) and that it is acknowledged that the core top age is uncertain. There are no Pb210 dates, or a surface core to correlate with. There should be a table with radiocarbon dates and paleointensity datums (depth, age, reference). It would summarize the information spread out over pages 9-10 and shown in Figure 3. I suggest bringing back Table 1 from the original submission, adding the magnetic datums, and addressing the original reviewer comments to this version.*

Reply: We agree that the core top in ARA 01-GC may not represent the modern age due to some sediment loss in the coring process. This is indicated by the absence of oxidized brown sediment at the core top, as opposed to a multi-corer collected at the same site. Nevertheless, we believe that the top of 01-GC is close to the sediment surface based on the biomarker distribution. Fig. 1 (attached below) is the concentration profile of IP25 and brassicasterol (Stein et al., 2017). We suppose that the downward decrease in concentrations of both compounds in the top 10 cm indicates their degradation with burial. A similar extent of brassicasterol concentration decrease occurs also in some of the deeper intervals, but is unique for the upper ~200 cm, while the IP25 decrease at the top is unique for the entire record.

We provided according explanations to this part and indicate that the core-top age is uncertain (Line 249 to 258). We brought back Table 1 with the paleomagnetic datums as supplementary table 2.

[Figure]

Fig. 1. Concentrations of brassicasterola nd IP25 in core 01A-GC (Stein et al., 2017).

*Divide section 3 in subsections: e.g. 3.1 Coring and Sampling, 3.2 Chronology, 3.3 XRD Mineralogy*

Reply: We divided section 3 into subsections 3.1. Coring and Sampling, 3.2. Chronology, 3.3. XRD mineralogy, as suggested.

*Figure 2 - From Panel E, one can see that there should be a data point with a CK/I ratio around 2.0 at about 63_N. This is not visible in Panel B. Check this carefully, as*

*there may be others? - At some sites, there are too many data points for this type of plot. An example: In Panel A, at the Mackenzie delta there are a lot of yellow dots, but they are covering up green ones as well. Either, make inserts for those areas, or make the dots smaller? - Panel E. The regression lines in CK/I and C/I vs latitude do not extend further south than 65N. Correct this or explain why.*

Reply: The symbol of the sample having a CK/I of 2.0 in the Yukon River estuary is hidden by another sample in Fig. 2B. Enlarged maps for Mackenzie and Yukon River estuary areas are put in supplementary material (Supplementary Figs. 1 and 2). The regression lines show the trend for the Chukchi Sea. This suffices to show a northward decrease of the ratios north of Bering Strait. The Bering Sea sediments do not show a systematic trend, probably reflecting multiple sources of chlorite, such as the Yukon River, Aleutian Island, etc. We added according explanations in the caption of Fig. 2.

*Figure 3. What do the crosses represent? Radiocarbon dates, paleointensity datums? Please specify. Add them all to a table (perhaps supplementary).*

Reply: Crosses represent radiocarbon dates in 01-GC and 5JPC and paleointensity datums in 06JPC. We added this information in the caption. All datums are shown in supplementary table 2.

*Figure 3. Rather than showing "D" for dolomite rich layers, please show the actual dolomite data. Also, add to the methods how dolomite was quantified (lines 250-260), and add the data to the supplementary tables.*

Reply: Dolomite intensity was added in Fig. 3, and the method was added to the text (lines 346 and 347). The data are presented in supplementary tables 4 and 5.

*Figure 3B. Please make it possible to distinguish between samples from the piston core vs trigger core by using different symbols.*

Reply: We showed open circle symbols for 05TC samples in Fig. 3B.

*Figure 4B. Same comment. Around 4000 cal yrs BP, there seem to be two data points for the same age. Is one JPC and one TC? The difference in their C/I values are large. Does this illustrate the uncertainty of the method?*

Reply: Both samples were derived from core 5JPC (392 and 398 cm). The difference in the values is larger than the analytical error. We assume that this difference could be related to a high-amplitude fluctuation that was observed at the same stratigraphic level in core 01-GC. We added an according explanation (Lines 430 to 433).

*Page 22 line 515. Correct "brassicasterol".*
Reply: This is corrected.

*Page 23 line 538. Add citation to Jakobsson et al 2017 Climate of the Past (this same special issue).*
Reply: Jakobsson et al. (2017) is cited.

Reply to referee #2 (Dr. T. M. Cronin) on "Holocene dynamics in the Bering Strait inflow to the Arctic and the Beaufort Gyre circulation based on sedimentary records from the Chukchi Sea" by Masanobu Yamamoto et al.

We thank Dr. Cronin for his helpful comments on our manuscript. Below is our reply to the main comments.

*Comment: This is a good paper on Holocene variability in a key part of the Arctic based on 3 sediment cores with decent chronology. The attached PDF has a number of comments inserted, including many minor problems with English.*

Reply: Thank you for recognizing the significance of our paper. We revised it according to your suggestions. Minor English problems were corrected as commented.

*But the most important problem with the paper is the confusing discussions in several places about the causes of variability in the mineralogical proxies used [if we accept the authors' ideas on what these proxies signify in terms of sea ice and ocean circulation]. Giving the benefit of the doubt on proxies, the paper should simplify mechanisms to explain patterns: long term insolation change during the Holocene, millennial-centennial TSI solar forcing, sea-level wind etc forcing Bering Strait inflow, the Arctic Oscillation affecting the Beaufort Gyre and Transpolar drift. Can these few mineralogical indices really distinguish among all these factors? Instead, can the authors highlight those patterns that are most important, like the shift in circulation near 1000 years ago. Or the early Holocene thermal warming. Or just the sea ice history? In the final revision, please make it easier for readers to see the main take-home messages and which hypotheses are supported.*

Reply: To simplify the explanation of the identified paleoceanographic changes, we have added the following table (Table 1) summarizing the patterns of the paleo-BG circulation and BSI, along with their possible forcings.

 Table 1. Summary of Holocene variability in the BG and BSI in northern Chkchi Sea

| Current system | Holocene trends | Multi-centennial to millennial cyclicity |
| --- | --- | --- |
| Beaufort Gyre (BG) circulation | Gradual weakening in response to decreasing summer insolation | ~0.36, 0.5, 1, and 2-kyr cycles paced by changes in solar activity |
| Bering Strait inflow (BSI) | Geographically variable. Mid-Holocene strengthening evident at the 01A-GC site, presumably due to weaker Aleutian Low | Geographically variable. ~0.36, 0.5, 1, and 2-kyr cycles paced by changes in solar activity are identifiable in 01A-GC |

*Line 84: Can you quantify the BSI in terms of its contribution in heat flow, relative to other ocean, atmospheric sources? Or just volume in Sverdrups compared to the other exchange routes into the Arctic? I guess some is covered below.*

Reply: We have added the following clarification in the introduction: "Mooring data suggest that an increase in the BSI volume by ~50% from 2001 (~0.7 Sv) to 2011 (~1.1 Sv) has driven an according increase in the heat flux from ~$3 \times 10^{20}$ J to ~$5 \times 10^{20}$ J (Woodgate et al., 2012)." (Lines 98 to 101)

*Line 222: Is there an alternative possible age model? The age for the base of the core is really important.*

Reply: At this point, no chronostratigraphic constraint is available for the lower part of the core, below the occurrence of material suitable for radiocarbon dating. Glaciomarine sediments were clearly deposited in sedimentological conditions different from those of the marine Holocene unit, which precludes the extrapolation of sedimentation rates derived from the $^{14}$C ages to the core bottom.

*Line 407: This is a huge conclusion, perhaps requiring more rigorous statistics and mechanistic explanation.*

Reply: We do not see anything unexpected or sensational in this conclusion. It is consistent with data from other Holocene studies (Hu et al., 2008; Anderson et al., 2005; Fisher et al., 2004; Sagawa et al., 2014), including the Chukchi shelf (Stein et al., 2017). We have revised the sentence to "This pattern suggests that millennial-scale variability in the BG was principally forced by changes in solar irradiance as the most likely forcing. Proxy records consistent with solar forcing were reported from a number of paleoclimatic archives, such as Chinese stalagmites (Hu et al., 2008), Yukon lake sediments (Anderson et al., 2005) and ice cores (Fisher et al., 2008), as well as marine sediments in the northwestern Pacific (Sagawa et al., 2014) and the Chukchi Sea (Stein et al., 2017)." (Lines 509 to 515)

*Line 485: check throughout the paper sea-ice versus sea ice [no hyphen] when used as an adjective.*
Reply: Corrected.

*Line 558: what is the island rule?*
Reply: The island rule is a concept used for modeling the direction and flow volume of an ocean current along the coast of an island or continent under a certain wind stress field (Godfrey, 1989). We, however, realize that the mention of the Island Rule is not necessary in this paper, so we have removed the phrase "based on the island rule (Godfrey, 1989)."

*Line 611: Can you make conclusions in bullet form? There is confusion about insolation, TSI-Solar forcing versus other processes in the BSI inflow. Also the AO mode of variability seems prominent, but no discussion of Pacific multidecadal PDO var.*
Reply: This section has been expanded to provide more explanation to the main conclusions, and a brief summary has been added in Table 1. We note that our records show multi-centennial and millennial-scale variability in the BG circulation and the BSI, which both seem to respond to changes in solar activity. To what extent the AO and PDO are involved in the BG and BSI dynamics is less clear and requires further investigation (see discussion in sections 5.1 and 5.6).

[revised manuscript text omitted]

Fig. 1

[Figure]

Fig. 2

[Figure]

Fig. 3

[Figure]

Fig. 4.

[Figure]

Fig. 5

[Figure]

Fig. 6

[Figure]

Fig. 7

[Figure]

Fig. 8